# Recent Research Trends in Neuroinflammatory and Neurodegenerative Disorders

**DOI:** 10.3390/cells13060511

**Published:** 2024-03-14

**Authors:** Jessica Cohen, Annette Mathew, Kirk D. Dourvetakis, Estella Sanchez-Guerrero, Rajendra P. Pangeni, Narasimman Gurusamy, Kristina K. Aenlle, Geeta Ravindran, Assma Twahir, Dylan Isler, Sara Rukmini Sosa-Garcia, Axel Llizo, Alison C. Bested, Theoharis C. Theoharides, Nancy G. Klimas, Duraisamy Kempuraj

**Affiliations:** 1Institute for Neuro-Immune Medicine, Dr. Kiran C. Patel College of Osteopathic Medicine, Nova Southeastern University, Ft. Lauderdale, FL 33328, USA; jc4962@mynsu.nova.edu (J.C.); am3707@mynsu.nova.edu (A.M.); kd1633@mynsu.nova.edu (K.D.D.); ysanche1@nova.edu (E.S.-G.); rpangeni@nova.edu (R.P.P.); kaenlle@nova.edu (K.K.A.); at1218@nova.edu (A.T.); ss3377@nova.edu (S.R.S.-G.); allizo@nova.edu (A.L.); abested@nova.edu (A.C.B.); ttheohar@nova.edu (T.C.T.); nklimas@nova.edu (N.G.K.); 2Department of Pharmaceutical Sciences, Barry and Judy Silverman College of Pharmacy, Nova Southeastern University, Ft. Lauderdale, FL 33328, USA; ngurusam@nova.edu; 3Miami VA Geriatric Research Education and Clinical Center (GRECC), Miami Veterans Affairs Healthcare System, Miami, FL 33125, USA; 4Cell Therapy Institute, Dr. Kiran C. Patel College of Allopathic Medicine, Nova Southeastern University, Ft. Lauderdale, FL 33328, USA; gravindr@nova.edu; 5Laboratory of Molecular Immunopharmacology and Drug Discovery, Department of Immunology, Tufts University School of Medicine, Boston, MA 02111, USA

**Keywords:** Alzheimer’s disease, Gulf War Illness, human induced pluripotent stem cells, myalgic encephalomyelitis/chronic fatigue syndrome, neuroinflammation, Parkinson’s disease, traumatic brain injury

## Abstract

Neuroinflammatory and neurodegenerative disorders including Alzheimer’s disease (AD), Parkinson’s disease (PD), traumatic brain injury (TBI) and Amyotrophic lateral sclerosis (ALS) are chronic major health disorders. The exact mechanism of the neuroimmune dysfunctions of these disease pathogeneses is currently not clearly understood. These disorders show dysregulated neuroimmune and inflammatory responses, including activation of neurons, glial cells, and neurovascular unit damage associated with excessive release of proinflammatory cytokines, chemokines, neurotoxic mediators, and infiltration of peripheral immune cells into the brain, as well as entry of inflammatory mediators through damaged neurovascular endothelial cells, blood–brain barrier and tight junction proteins. Activation of glial cells and immune cells leads to the release of many inflammatory and neurotoxic molecules that cause neuroinflammation and neurodegeneration. Gulf War Illness (GWI) and myalgic encephalomyelitis/chronic fatigue syndrome (ME/CFS) are chronic disorders that are also associated with neuroimmune dysfunctions. Currently, there are no effective disease-modifying therapeutic options available for these diseases. Human induced pluripotent stem cell (iPSC)-derived neurons, astrocytes, microglia, endothelial cells and pericytes are currently used for many disease models for drug discovery. This review highlights certain recent trends in neuroinflammatory responses and iPSC-derived brain cell applications in neuroinflammatory disorders.

## 1. Introduction

Alzheimer’s disease (AD), Parkinson’s disease (PD), traumatic brain injury (TBI) and Amyotrophic lateral sclerosis (ALS) are chronic neuroinflammatory and neurodegenerative disorders [1]. Neurodegenerative disorders are characterized by the presence of dysfunction of synapses, neurons, glial cells and network connections [2]. Gulf War Illness (GWI) and Myalgic encephalomyelitis/chronic fatigue syndrome (ME/CFS) are chronic disorders that are associated with some neuroimmune dysfunctions without apparent neurodegeneration [3]. Neurodegenerative diseases cause a major health challenge as more than 50 million people suffer from these disorders [4]. Neuroinflammation is the response of the central nervous system (CNS) to injury, toxicity, infection and disease [5]. Neuroinflammation may be beneficial or detrimental to the brain. Neuroinflammation is characterized by increased levels of inflammatory molecules in the CNS that could cause progressive neurodegeneration, functional impairment [6], glial cell activation and brain damage [3,7]. Chronic CNS inflammation is mediated by proinflammatory cytokine-activated microglial cells and reactive astrocytes, resulting in neural damage and dysfunction of the blood–brain barrier (BBB) [8,9]. Imbalance of anti-inflammatory and proinflammatory molecules induces neuroinflammation and hyperactivity of glial cells [10]. Understanding neuro-immune interactions offers insight into the underlying cellular and molecular mechanisms of inflammation in neurodegeneration and potential therapeutic interventions [11,12]. The precise mechanism of disease pathogenesis of these chronic neuroimmune disorders is currently not known. Neuroinflammatory responses play an important role in neurodegenerative disorders. Glial cells, especially microglia and astrocytes, are the important regulators of neuroinflammation in the brain. These glial cells are categorized as neurotoxic/proinflammatory M1 microglia and A1 astrocytes, as well as anti-inflammatory/neuroprotective M2 microglia and A2 astrocytes [13]. A1 astrocytes secrete and release several proinflammatory mediators and neurotoxins, but A2 astrocytes release neurotrophic mediators that support neuronal cell growth [14]. The phenotypic status of these cells can change, based on the progression and severity of neuroinflammatory and neurodegenerative diseases [13]. Chronic neuroinflammation induces the generation of proinflammatory M1 microglia and A1 astrocytes [10]. The extent of neuroinflammation and neurodegeneration is regulated by genetic variations in the CNS and peripheral immune cells [12]. Chronic neuroinflammation can affect specific neuronal types in specific regions and induce specific neuroinflammatory and neurodegenerative disorders [12]. Mast cells release inflammatory molecules that can activate glia and induce neuroinflammation in neuroinflammatory disorders [15,16]. Microbial dysbiosis, leaky gut, and cerebral and vascular barriers are implicated in neurodegenerative disorders [17]. Peripheral-derived inflammatory mediators and brain-derived inflammatory cytokines, chemokines, α-synuclein, corticotropin-releasing hormone (CRH), substance P (SP), beta-amyloid 1-42 (Aβ1-42) peptide and amyloid precursor proteins (APP) can activate glial cells and immune cells in the brain, and cause further release of inflammatory and neurotoxic mediators enhance chronic neuroinflammation and neurodegeneration in the brain [15]. Glial-cell-derived mediators including interleukin-1 beta (IL-1β), IL-6, IL-8, IL-33, chemokine(C-C motif) ligand 2 (CCL2), tumor necrosis factor-alpha (TNF-α), reactive oxygen species (ROS), nitric oxide (NO), matrix metalloproteinases (MMPs), vascular endothelial growth factor (VEGF) and other mediators induces neuroinflammation [15].

Human induced pluripotent stem cell (hiPSC)-derived neurons, astrocytes, microglia, endothelial cells, and pericytes are currently used for many diseases modeling drug discovery for various neuroinflammatory disorders [18]. These human hiPSC-derived cells are used for various co-culture systems, brain organoids, and neurovascular unit cultures [19,20]. There are no disease-modifying treatment options currently available for neuroinflammatory disorders and are treated symptomatically. Moreover, drug delivery to the CNS remains a challenge in treating neurodegenerative disorders due to the BBB [21,22]. Available drugs address only symptomatically. However, newer innovative techniques such as nanoparticles, exosomes and intranasal administration could be useful [4,21]. This review highlights certain trends in disorders associated with neuroimmune and inflammatory responses, as well as the use of iPSC-derived brain cells in neuroinflammatory mechanisms and neurotherapeutics.

## 2. Alzheimer’s Disease (AD)

AD, the most common cause of dementia, constitutes about 60–70% of all cases of dementia worldwide and is an age-related neurodegenerative disease characterized by progressive memory loss. AD is one of the major public health problems in the world [4]. Over 6 million people in the United States have Alzheimer’s dementia. By 2050, this number is projected to increase to about 13 million (Alzheimer’s Association) [23]. It is estimated that more than 360,000 new cases occur in the United States each year. The pathophysiology of AD is still not completely understood, but involves the deposition of amyloid plaques (APs) containing Aβ and neurofibrillary tangles (NFTs) due to hyper-phosphorylated tau protein, and glial activation-associated neuroinflammation with neuronal loss in the brain [4,14,24]. AD pathogenesis involves Aβ abnormality, tau phosphorylation, neuroinflammation, neurotransmitter dysregulation, and enhanced oxidative stress [25]. Figure 1 depicts the neuroinflammatory processes in the pathogenesis of AD.

Hallmarks of AD are APs and NFTs [24]. The process by which neurons are killed in AD is not known. However, necroptosis, parthanatos, ferroptosis, and cuproptosis are proposed, and these need further studies [26]. Coronavirus disease 2019 (COVID-19) is a risk factor, and can contribute to the pathogenesis of AD [27]. Brain injury could accelerate the pathogenesis of AD in high-risk people [28]. Abnormal levels of beta-amyloid protein in an AD patient will self-aggregate and form extracellular plaques that are toxic and affect the brain structure and function [29]. Amyloid-beta can further aggregate and deposit into the vasculature of the brain and cause cerebral amyloid angiopathy (CAA). Further, hyperphosphorylated tau proteins can twist around each other and accumulate inside neurons to form NFT, which will further damage the cellular structure and function of the brain [30]. The presence of both Aβ and phosphorylated tau proteins, along with other critical cellular and molecular processes, is associated with a higher probability of AD development [31]. Due to the fact that these brain pathologies and biomarkers appear much earlier than the manifestation of clinical symptoms, and early detection is vital for efficient intervention. Enhanced oxidative stress contributes to the pathogenesis of AD [32].

Astrocytes are the most abundant glial cells in the brain, and play an important role in the pathophysiology of AD. In healthy individuals, these cells play an important role in maintaining the BBB, synaptic remodeling, regulating neuroinflammation, and ion homeostasis [13]. Astrocytes also differentiate into a pro-inflammatory phenotype A1 or neuroprotective/anti-inflammatory phenotype A2 [13]. In AD patients, cross-sectional and longitudinal studies found that A1 astrocytes, which produce inflammatory complement proteins of both the classical and alternative complement pathways in astrocyte-derived exosomes (ADEs), are increased in AD patients compared to controls [33]. Chitinase-3-like protein (CHI3L1/YKL-40), an A1 astrocytic protein, has been identified as another potential cerebrospinal fluid (CSF) biomarker, which increases with aging and early in AD [34]. Moreover, a study proposed how CHI3L1 could potentially contribute to the comprised BBB integrity seen in early AD, loss of synapses and other neurodegenerative processes, giving this biomarker more significance [35]. One cross-sectional study compared patients with early- and late-onset AD, and found that glial fibrillary acidic protein (GFAP) was elevated in both groups of AD patients compared to controls, suggesting its use as a marker in AD and confirming astrocyte activation and degeneration as a part of the pathology of the disease [36]. The neuronal axonal damage marker neurofilament light (NfL) chain and degenerative composite GFAP and NfL are elevated in AD [36]. Another study suggested that significantly increased plasma GFAP level as an early marker for brain Aβ pathology and astrocytosis, but not tau aggregation in AD pathogenesis [37]. Advances in molecular imaging and other techniques have enabled further detection of more specific AD markers, such as Colony-Stimulating Factor 1 Receptor (CSF1R) and P2Y12 receptor. CSFR1 is mostly expressed in microglial cells in the brain, and contributes to microglial growth, proliferation and survival [14,38]. Upregulation of this receptor has been shown to parallel with neuropathology in AD. P2Y12 receptors are also microglial receptors that enable the cells to monitor neuronal function. Immunohistochemical staining demonstrated decreased levels of the P2Y12 receptors in the brains of AD patients and taupathy mice model [14,39]. P2Y12R is an important PET biomarker currently explored for microglia [40]. A different approach focuses on the study of alterations in glucose metabolism and glucose uptake in the brains of AD patients and the effects on concentrations of certain metabolites. Specifically, reduced glucose metabolism suppressed the metabolism of L-serine, and thus D-serine, which plays a role in synaptic potentiation. Impairment of glycolysis-derived L-serine release from astrocytes induced cognitive disorders in AD [41]. Therefore, astrocytic glycolysis could control cognitive functions and oral L-serine could be a therapeutic option in AD [41]. Glia maturation factor (GMF) is a proinflammatory mediator that plays an important role in the pathogenesis of AD by activating microglia, astrocytes and mast cells to release several proinflammatory and neurotoxic mediators [15,42]. Several pathways, such as nuclear factor-kB (NF-kB), mitogen-activated protein kinase (MAPKs), and Akt/mammalian target of rapamycin (mTOR), are involved in the pathogenesis of AD [10].

Several therapeutic options have been evaluated for many decades, but still, there is no cure for AD [4]. However, there are treatments available to manage and help slow down the progression of the disease. Attempts to treat AD using anti-amyloid strategies did not provide significant benefits [43]. Thus, in addition to the “amyloid cascade hypothesis”, additional factors such as neuroinflammation could play an important role in neurodegenerative diseases [43]. Thus, there are different types of mechanisms that may be involved independently or simultaneously in neuronal loss in AD [26]. Some of the earlier approved treatments include cholinesterase inhibitors, such as Donepezil and memantine, an N-methyl-D-aspartate (NDMA) receptor antagonist. Cholinesterase inhibitors work by increasing the neurotransmitter acetylcholine (Ach) in the brain to address the deficiency in AD patients and thus help in the transmission of information between neurons [4]. Further, amyloid-beta plaques contribute to hypersensitivity and excitotoxicity of NDMA receptors, a glutamate receptor that plays a key role in synaptic plasticity [44]. These treatments help manage the side effects of the disease rather than treat the cause. In fact, it was shown that combination therapy of cholinesterase inhibitors and memantine had little success, indicating that efficient treatment for AD should target multiple pathways of the disease pathogenesis [44]. These findings demonstrate that, although there are increasing technologies in the detection of biomarkers in AD, there is still a need for more disease-specific markers that can be used for early detection and as targets for drug development.

## 3. Parkinson’s Disease (PD)

PD is a chronic progressive multisymptomatic neurodegenerative movement disorder characterized by neuroinflammation and dopaminergic neurodegeneration associated with substantia nigra pars compacta of the brain [45], accumulation of alpha-synuclein in the dopaminergic neurons, Lewy bodies formation, and neurodegeneration in the brain [46]. About one million people in the United States are affected by PD [47]. This could increase to 1.2 million by 2030. About 90,000 people in the US are diagnosed with PD each year [47]. PD is the second most common neurodegenerative disease after AD [47]. The molecular mechanism and the trigger in the pathogenesis of PD are not yet clearly understood [48]. Many PD patients show cognitive impairment due to coexisting α-synuclein and AD pathologies [49]. The structure and functioning of the brain are further interrupted by the buildup of the alpha-synuclein protein. As a result of this accumulation of proteins within the brain, dementia can manifest and present with the further progression of PD. Other risk factors that contribute to the development of PD include advanced age, longevity, and industrial by-products and decreasing smoking rates [50]. Additionally, being a male is another risk factor for PD [51]. Environmental factors, chemicals and genetic factors are also implicated in the pathogenesis of PD [52]. PD diagnosis can be made based on the history of the patient, clinical physical exam and a magnetic resonance imaging (MRI) of the brain. While these factors may tip clinicians toward a diagnosis of PD, there is no one diagnostic study of choice for this progressive neurological disease. A recent study indicated that α-synuclein seed amplification assay (SSA) is a useful diagnostic tool for distinguishing between different types of PD [53].

Neuroinflammation can induce and accelerate the pathogenesis of PD [15]. PD-relevant toxin 1-Methyl-4-phenylpyridinium (MPP+), a metabolite of parkinsonian neurotoxin 1-methyl-4-phenyl-1,2,3,6-tetrahydropyridine (MPTP), stimulates mast cells to release inflammatory mediators that can also activate and release inflammatory mediators from glia-neurons that are implicated in PD pathogenesis [46,54]. Mast cell activation could play a role in neuroinflammation in PD [55]. Inflammation and stress can exacerbate PD in those who may be genetically predisposed. Neurotropic pathogens including viruses may enter the nervous system through the nasopharynx and intestinal mucosa, which can lead to degeneration within the substantia nigra pars compacta [56]. Pathogens may also lead to an oxidative state that results in neuronal damage and accumulation of proteins within the brain. In addition to the buildup of these proteins, it is hypothesized that microglia may exacerbate the buildup and, if not, “spread” the disrupted proteins from one cell to another [48]. Finally, conditions associated with inflammation were found to increase the risk of developing PD. For example, individuals with Type 2 diabetes mellitus have mitochondrial disruption and increased risk of PD [57,58]. Studies have also suggested that impairment in autophagy can precipitate PD due to allowing malformed proteins and Lewy bodies to accumulate and contribute to the progression of PD [59].

Another study evaluated if there was a genetic locus that influenced the age of onset of Parkinson’s [60]. In this genome-wide associated study, it was found that the *SNCA*, *TMEM175*, and glucocerebrosidase (GBA) genes were associated with the alpha-synuclein mechanisms of PD pathogenesis [60]. *SNCA* encodes for the alpha-synuclein protein, which is a large portion of the Lewy bodies. Duplications or triplications of this gene were associated with an increased risk of development of PD. *TMEM175* was associated with worsened lysosomal and mitochondrial functions, possibly contributing to the buildup of alpha-synuclein [60]. *GBA* is a gene that codes for lysosomal transfer, and the defect is involved in alpha-synuclein aggregation [60]. These genes play a significant role in the development of PD; however, not all these loci made a significant difference in the age of onset. Blood and neuronal extracellular vesicle biomarkers may be useful in determining cognitive prognosis in PD [49].

## 4. Traumatic Brain Injury (TBI)

TBI is defined as “an alteration in brain function, or other evidence of brain pathology, caused by an external force or trauma. Traumatic impact injuries can be defined as closed (or non-penetrating) or open (penetrating)” [61]. There are more than 5.3 million individuals with permanent brain injury-associated disability living in the United States [61]. About 2.8 million Americans sustain TBI in the United States every year [61]. Figure 2 depicts the neuroinflammatory processes in the pathogenesis of TBI. TBI could be due to sports injuries, motor vehicle accidents, falls, assaults, military accidents, blast injuries from improvised explosive devices (IED), and gunshots [28]. About 19.5% of soldiers deployed in Operation Iraqi Freedom (OIF) and Operation Enduring Freedom (OEF) were subject to blast TBI (bTBI) [28]. TBI is a risk factor for PD and AD [62]. TBI could be classified into mild, moderate and severe types [28]. This multifaceted condition is characterized by primary and secondary injury, each triggering a cascade of neurological alterations, both immediate and prolonged [28]. TBI induces immune responses to protect the brain from any additional damage. However, an excessive and continued immune response can lead to neuroinflammation-mediated neurodegeneration leading to AD [63]. Primary TBI inflicts localized brain impairment, while secondary TBI initiates a complex neuroinflammatory sequence leading to the compromise of the BBB, cerebral edema, the release of diverse immune mediators, and the potential death of neuronal [64,65].

Several mechanisms contributing to secondary TBI include excitotoxicity, mitochondrial dysfunction, oxidative stress, lipid peroxidation, neuroinflammation, axonal degeneration, and eventually cell death [64]. Various biomarkers have been linked to TBI, and several are under consideration for standardization and implementation in clinical practice, limited by the challenges of the heterogeneous nature of TBI and the variable course of pathology [66]. Some commonly referenced biomarkers associated with secondary TBI include NFL chain, ubiquitin carboxy-terminal hydrolase-L1 (UCH-L1), tau, S100B, and GFAP [67]. The intricate interplay of cytokine release, oxidative stress, and the recruitment of immune cells substantially contributes to the enduring consequences of secondary TBI.

Mechanical injury sustained in TBI results in diffuse axonal injury, characterized by degradation of axonal cytoskeletal components and disruption of axonal transport, resulting in swelling, apoptosis, and degradation of the myelin sheath [64]. Degenerative alterations significantly correlate with cognitive outcomes, even in cases characterized as mild TBI [67]. Diffuse axonal injury is associated with increased levels of biomarkers within the CSF and serum such as NfL, myelin basic protein (MBP), GFAP, and tau protein, among others [62,68,69,70]. Initial depression of cellular excitability immediately post-injury may be observed clinically as hyporeflexia, hypotonicity, and sensory loss due to severing axonal tracts, neuronal cell death, and loss of pre-synaptic input [71]. Excitotoxicity is a central mediator of TBI pathophysiology, primarily mediated by the excessive release of excitatory neurotransmitters, most notably glutamate [64,72,73]. Increased permeability of BBB following TBI is a driver for the increased permeability of excitatory neurotransmitters. Once bound to its receptors, most notably N-methyl-D-aspartate receptor (NMDAR) and α-amino-3-hydroxy-5-methyl-4-isoxazolepropionic acid receptor (AMPAR) activation, an influx of calcium ions is responsible for triggering downstream cellular signaling, culminating in cell death [64]. This abrupt influx of calcium is also associated with mitochondrial dysfunction, leading to the formation of ROS and impaired adenosine triphosphate (ATP) synthesis. Increased free radical production, particularly through lipid peroxidation, results in cellular damage, impaired cerebral blood flow, and diminished brain plasticity. Molecular pathways involving proteases such as caspases contribute to apoptosis, resulting in compromised cellular functions and ultimately cell death [64,72]. Diffusion Tensor Imaging (DTI) may be used to assess white matter damage resulting from diffuse axonal injury (DAI) [74]. Axonal injury, in addition to the resultant acute and chronic neuroinflammatory state, is critical, and may serve as targets for identifying potential therapeutic interventions and targets to mitigate the long-term consequences of TBI [28,67,74].

The disruption of the BBB is mediated by several cellular and molecular mechanisms, and contributes to the initiation of neuroinflammation characteristic of secondary TBI [73,75]. Neutrophils, macrophages, microglia, T-cells, and mast cells impact the inflammatory state following TBI [76]. Inflammatory cytokines implicated include IL-1, IL-6, TNF-α and chemokines, which contribute to BBB permeability, glial activation, and immune cell recruitment following TBI [28]. Activation of brain endothelial cells triggers the release of cytokines, chemokines, and activation of microglia of the CNS, resulting in recruitment of other immune pathways. Activation of Rho/Rock, protein kinase C (PKC), and mitogen-activated protein kinase (MAPK) pathways via CNS microglia alters tight junction distribution and endothelial permeability, allowing an influx of immune cells and peripheral proteins [75]. Mechanical and molecular mechanisms can alter endothelial integrity, causing upregulation of molecules such as intercellular adhesion molecule-1 (ICAM-1) and modulation of Major Facilitator Superfamily Domain Containing Protein-2a (Mfsd2a) [75,77]. Astrocytes are important for the integrity of the BBB due to their communication with endothelial cells [28]. A vicious cycle of increased permeability can be induced by alterations in this relationship through changes in aquaporin 4 expression and distribution, infiltration of the CNS by peripheral proteins such as albumin, the release of MMPs which degrade the basement membrane, VEGF, endothelin-1, and glutamate release, each of which leads to further increases BBB permeability, resulting in edema and secondary brain injury [73,75]. Mast cell activation mediated multifactorial inflammatory mediator release can induce neuroinflammatory response and neurodegeneration in TBI [28].

TBI induces an inflammatory response involving the activation of microglia and release of cytokines and chemokines, as well as increased permeability of the BBB [64,72]. Under normal circumstances, T-cells and mast cells monitor meningeal spaces and the BBB for substances foreign to the CNS. However, following brain injury chemokines upregulate adhesion molecules on endothelial cells, activate microglial cells, and trigger an influx of T-cells. The elicited immune response’s primary objectives are to clear cellular debris and enhance wound healing [28]. These processes, while essential for repair, can lead to detrimental effects under conditions of chronic inflammation. A vicious positive feedback loop can result in persistent inflammation and immune upregulation, which results in secondary brain injury [28,64,72]. A prolonged state of neuroinflammation following TBI is associated with structural changes and neurodegeneration. Axonal injury, accumulation of Aβ plaque, tau pathology and chronic inflammation may contribute to the risk of development of neurodegenerative diseases following TBI [28,67,74,78]. Studies have identified several biomarkers associated with the immune response in TBI. Inflammatory biomarkers such as interleukins (IL-1, IL-6, IL-8, IL-10, TNF-α), adiponectin, high mobility group box 1 (HMGB1), galectin-3, Ficolin 3, and mannose-binding lectins (MBL) are elevated after TBI [62]. Their correlation with injury severity and outcomes varies and remains under study [62]. Elevated IL-6 and IL-8 levels in blood and CSF are associated with fatal TBI and cerebral hypoperfusion. MBL, Galectin-3, and Ficolin 3 are associated with TBI severity and outcomes. Adiponectin and HMGB1 are correlated with increased mortality and poor outcomes following TBI [79].

NfL is a protein specific to neuronal cells that provides structural integrity to the neuronal cytoskeleton and is a recognized biomarker used to evaluate axonal damage and neurodegeneration. NfL can be elevated in non-traumatic neurodegenerative diseases such as AD, dementia, stroke, and others [28,67,70,80]. Elevated levels of NfL in the context of TBI reflect more severe or widespread axonal damage, and are associated with cognitive decline [68,78,81,82], as well as poor outcomes and increased mortality [62,69,81]. Further inquiry into reliable clinical thresholds remains to be determined, as well as standardization of when to apply this biomarker in the course of pathology following TBI [70,83]. NfL requires further exploration of its precise clinical utility, but serves as a promising biomarker for the diagnosis and prognosis of TBI [80]. MBP is the second major protein within the CNS, more specifically found within oligodendroglia. Damage sustained following TBI results in elevated MBP levels and reflects demyelination following TBI [78]. Released into blood 1–3 days after injury, with an association between elevated MBP and increased risk of mortality. A limitation of MBP is its presence in both central and peripheral nervous tissue, therefore rendering MBP an inaccurate diagnostic or prognostic biomarker when evaluated in isolation [62]. Ubiquitin carboxy-terminal hydrolase-L1 (UCHL-1) is an enzyme found within neuronal cells and is considered a biomarker of interest following TBI. Increased UCH-L1 concentration in serum and CSF within 6-24 h post-TBI correlates with injury severity, clinical outcomes, and mortality [62,66]. Blood levels of UCH-L1 and GFAP are Food and Drug Administration (FDA)-approved biomarkers for mild TBI (mTBI) [84]. UCH-L1 and GFAP are markers for severe TBI, as well as in milder cases of mTBI and athletic concussions [68,78,85]. GFAP levels correlated with severe TBI and were predictive of low cerebral perfusion pressure (CPP), elevated intracranial pressure (ICP), and mortality in severe TBI. Several potential biomarkers of disruption of BBB include markers such as S100B and GFAP. S100B is a protein found within astrocytes, while GFAP is an intermediate filament protein present in astrocyte’s cytoskeletal structure. Elevations in either are associated with astroglia damage and disruption of the BBB following TBI [66]. Micro RNAs (MiRNA) have been recently for their role in the regulation of the BBB, via the regulation of genes encoding tight and adherens junction proteins, membrane transporters, and signaling pathways. Specific families of miRNA are essential for the regulation of angiogenesis, neuronal migration, and are critical following CNS injury [65].

Neuron-specific enolase (NSE) is an enzyme present primarily in neurons and involved with glycolysis. However, it is a non-specific marker, as well as being non-specific to traumatic injury within neuronal tissues [68], but elevated levels of NSE were associated with neuronal damage following brain injury, in both mTBI and severe cases of TBI [66,78]. Tau is a microtubule-associated protein found in neurons implicated in axonal injury and neurodegenerative conditions both in association with TBI and non-traumatic degeneration of the brain [69]. Acute hyperphosphorylation of tau protein can be induced following TBI, resulting in the formation of NFTs [68]. This pathology may progress and contribute to the risk of neurodegenerative disorders such as AD [67,78]. Tau protein levels correlated with TBI severity and poor outcomes and, therefore, may be useful in diagnosing acute TBI and in chronic traumatic encephalopathy (CTE) [62,69]. S100B is a protein belonging to a family of calcium-binding proteins, primarily found in cells of the CNS, in particular astrocytes. It was the first biomarker used in Europe for concussions; however, it is not significantly specific or diagnostic in isolation [80]. Serum levels of S100B within 12–36 h post-TBI correlated with outcomes in neurointensive care units and were linked to Glasgow Outcome Score (GOS) at 6 months following injury. S100B evaluation in combination with GFAP and UCH-L1 to predict outcomes and mortality [62,66,68,69,78]. GFAP is an intermediate filament protein found primarily in the cytoskeleton of astrocytes. Like other structural protein markers, elevated GFAP is associated with astroglia damage and BBB disruption following TBI. Studies have shown an observable rise within the first 20 h post TBI and remained detectable for up to 7 days [66,78]. This can aid in evaluating the need for neuroimaging such as CT or MRI [62,69].

Extracellular vesicles (EVs), including exosomes and microvesicles, play a role in intercellular communication and contain markers indicative of glial cells and neurons’ state during CNS injuries, making them potentially useful biomarkers following TBI. These vesicles contain other biomarkers such as spectrin break-down products (SBDP), UCH-L1 and GFAP. The isolation of these EVs presents a technical challenge and great translatability [62], but shows promise in TBI diagnosis and prognosis due to their association with TBI severity and long-term outcomes [62,78]. MicroRNAs (miRNAs) regulate gene expression post-transcriptionally application [62]. Alterations in miRNA concentrations detected in serum, plasma and CSF following TBI, such as miR-92a, miR-320c, and miR-30, exhibit potential utility as diagnostic and severity assessment biomarkers for TBI [65]. Hormonal changes post-TBI reflect a disrupted endocrine system and may be predictive for patient outcomes including levels of plasma copeptin, nesfatin-1, and resistin [79]. A recent paper reported that IL-1 receptor antagonists as therapy for TBI [86]. Recent reports indicate that hyperbaric oxygen therapy (HBOT) was safe and beneficial in treating TBI including chronic sequelae of TBI [87,88].

## 5. Gulf War Illness (GWI)

GWI is a chronic complex and poorly understood multi-symptom disorder that affects about 30% of 700,000 veterans of the 1990–1991 Gulf War, with multiple symptoms that include fatigue, cognitive problems, headache, chronic pain, gastrointestinal and dermatologic disorders [89,90,91]. GWI is considered a neuroimmune disease and, currently, there is no cure for GWI [92,93]. GWI is most likely caused by exposure to toxic chemicals such as neurotoxins, pyridostigmine bromide (PB), and pathogens leading to environmental modifications of genetic profiles and xenobiotic metabolism in veterans [94,95,96,97]. Many symptoms of GWI are similar to those of autoimmune diseases, because the presence of autoantibodies against GFAP, MBP and tau has been reported in veterans with GWI than in controls [98]. However, the pathogenesis is different, as GWI is associated with hazardous chemical and pesticide exposures. The first direct evidence of brain upregulation of the neuroinflammatory marker 18 kDa translocator protein (TSPO) in veterans with GWI and supports neuroinflammation as a therapeutic target for this disorder [99]. Neuroinflammation and cognitive dysfunction are implicated in GWI [100]. Chronic neuroinflammation without apparent neurodegeneration is associated with systemic inflammation in GWI [3,90]. GWI veterans show increased levels of IL-1β, IL-6, IL-2, IL-10, IFN-γ, IL-4, IL-5, IL-17A, IL-33 and decreased level of IL-13 from T cells [98]. BBB dysfunction and increased MMP activity at the BBB could allow entry of peripheral immune cells into the brain in GWI [98]. Reliable blood biomarkers need to be discovered for GWI [90]. GWI pathogenesis may involve neuroinflammation, oxidative stress, and neuronal damage leading to disruption of neuronal function [92,101]. These alterations often are associated with a leaky gut, leading to increasing neuroinflammatory responses in affected individuals [102]. Immune responses from macrophages, B cells, cytotoxic T cells, helper T cells, and memory T cells are reported in GWI [98]. Veterans with GWI show higher levels of lymphocytes, monocytes and B cells, and CD4^+^/CD8^+^ ratio [98].

One study with GWI mouse models has shown that alteration in the virome signature implicated intestinal tight junction proteins, activation of innate immune mechanisms and Toll-like receptor (TLR) signaling pathways, increased serum cytokines such as IL-6, IL-1β, and IFN-γ, intestinal inflammation, and reduced neurogenesis marker brain-derived neurotrophic factor (BDNF) [103]. This study reported that enteric viral dysbiosis may activate enteric viral particle-mediated innate immune response in GWI [103]. Similarly, a decrease in butyrogenic and immune health-restoring bacteria in mouse models fed with a Western diet was associated with increased IL-6, claudin 2, and IL-1β, activated microglia, decreased BDNF, and increased levels of phosphorylated tau, which is an indicator of higher risk of cognitive deficiencies due to neuroinflammation [104]. TNF-α, IL6, CCL2, IL-1β, leukemia inhibitory factor (LIF), and oncostatin M denotes the neuroinflammatory responses in the brain in GWI [105,106]. Further, altered immune activity in GWI is associated with dysregulation in cellular energetics related to mitochondrial function and abnormal processing of pain and sensory stimuli in the brain [107,108]. Moreover, Ribonucleic acid sequencing (RNAseq), reduced representation bisulfite sequencing (RRBS) and H3K27ac ChiP seq have revealed epigenetic alterations such as histone modification and deoxyribonucleic acid (DNA) methylation changes in genes associated with neuroinflammation and cognitive symptoms [109]. Similarly, a signature of eight genes (seven downregulated, and one upregulated gene) was reported to be associated with neuroinflammation, other responses and signaling pathways related to the nervous system using differential pyridostigmine dosing in chronic GWI mouse model [110]. Biomarkers of neuroinflammation could be detected in the neuron- and astrocyte-derived extracellular EVs in GWI [100]. Inflammatory proteins can enter the blood through nanosized EVs from brain cells. Smaller EVs from brain cells can cross BBB and enter into the blood [100].

Based on GWI pathogenicity, the compounds that help to modulate neurological activities could be promising therapeutic targets for the disease. Recently, several compounds, such as polyphenol curcumin, monosodium luminol (MSL) and Minocycline, have shown a positive effect by upregulating antioxidant genes, alleviating neurogenesis and neuroinflammation, which showed improvement in cognitive function including reinstatement of redox homeostasis improvement in cognitive function, including reinstatement of redox homeostasis [111,112,113,114]. Further, antioxidant co-enzyme Q10 (CoQ10), a low-risk therapeutic agent, Bacopa monnieri, anti-inflammatory flavonoid concord grade juice (CGJ), and mifepristone therapies beyond Phase II clinical trials for GWI patients have shown some progresses for therapeutic interventions in addition to cognitive and behavioral therapies in place [115,116,117]. In addition, other randomized trials using Tai Chi, problem-solving treatment (PST), yoga and cognitive behavioral therapy (CBT) have been carried out to investigate their efficacies in cognitive performance of GWI patients [118,119,120,121,122]. These encouraging reports show that we need further investigations to have effective GWI therapies in practice. Despite undergoing clinical trials, and having some promise, there is still a lack of effective minimally invasive markers for the GWI and its neuroinflammation. Brain-derived EVs in blood-based liquid biopsy, however, could be a potential biomarker for GWI [90]. Therefore, we still need biomarkers and more effective and specific therapies for GWI.

## 6. Myalgic Encephalomyelitis/Chronic Fatigue Syndrome

ME/CFS is a complex, heterogeneous, and chronic neuroimmune disorder. The pathogenesis of ME/CFS is currently not known, but several factors including chemical exposures and infectious agents cause this condition [3]. ME/CFS and GWI show many symptoms such as fatigue, pain and cognitive disorders that are not controlled by rest [123]. Neurological symptoms of ME/CFS include muscle fatigue, mental fatigue, diminished cognition, psychomotor slowing, disrupted sleep, hypersensitivities to noise, light, and smells, headache, pain, paresthesia and extreme dysautonomia [124]. The unbearable physical and mental fatigue can remain in the ME/CFS patients for decades. About 836,000 to 2.5 million Americans have been reported to have ME/CFS, most common in people between 40 and 60 years old, where females appear to be more affected than men at a 4:1 ratio (Centers for Disease Control and Prevention (CDC) [125]. Both physiological and psychological stressors are implicated in the pathogenesis and severity of ME/CFS [3]. As the diagnosis of ME/CFS is difficult, another million may have this condition [3]. The “Three Pillars” hypothesis of ME/CFS pathogenesis [126] might be described as (a) dyshomeostasis of the CNS, (b) immune system, and (c) endocrine system. A disrupted BBB could potentially lead to chronic neuroinflammation observed in ME/CFS [127]. Neurological disruptions in ME/CSF include memory problems, headaches, sleep disturbances, unexplained joint and muscle pain, hypoactivation of the hypothalamic-pituitary-adrenal (HPA) axis and the basal ganglia, as well as unbalanced serotonin cycle, in addition to observed irregularities in the brain stem during brain imaging [128]. Presence of chronic neuroinflammation without apparent neurodegeneration and CNS pathology could occur in ME/CFS [3]. Neuroinflammation is present in the limbic system, midbrain and pons region of the brainstem [129]. Sleep disorders have been associated with damaged reticular formation, especially the reticular activating system (RAS) nuclei [130]. Hypocapnia and hyperventilation are also frequently associated with ME/CFS [124]. The post-exertional malaise (PEM) and orthostatic challenges in ME/CFS correlate with both diminished oxygen supply in muscles and low cerebral blood flow [131]. Furthermore, elevated levels of stress hormones, such as cortisol, may have immunosuppressive effects, acting on neutrophils and natural killer cells [132], contributing to the development of ME/CFS. Because the brain is unable to control stress in ME/CFS in the paraventricular nucleus (PVN) of the hypothalamus, the stress might be a constant stimulus of an immune response in the CNS, to trigger relapses and partial recovery cycles [133].

Infective agents induce B cell immune responses in ME/CFS patients [134]. Impairment of ATP synthesis by dysfunctional mitochondria is associated with the upregulation of mitochondrial complexes and defective production of ROS and coenzyme A [135], evident in ME/CFS relapses. Circadian rhythms are disrupted in ME/CFS [136]. Post-infection state (virus, bacteria *Coxiella burnetii*.; protozoa *Blastocystis* spp. or *Dientamoeba fragilis*) [137] is known to trigger a cascade of cytokine responses that ultimately contribute to ME/CFS pathology. Similarly, this is observed in bacterial and viral reactivation [138], in which reactivated viruses (Human Herpesvirus Type-6 (HHV6), Epstein Bar Virus (EBV), Herpes Simplex Virus-1 (HSV-1, a neurotropic virus)) greatly influence morbidity through a dysregulated immune system and inflammation. During dormancy periods [139], and the IgG levels against HSV-1 are associated with brain fog, AD, schizophrenia, and bipolar disorders [140]. HHV-6 has also been linked to the development of encephalopathy, ME/CFS and other neurological disorders [141], and is known to infect CD4^+^ T-cells preferably, but also CD8^+^, macrophages/monocytes and natural killer (NK) cells. HHV-6-infected cells affect immune responses [142]. EBV infection is known to trigger the development of ME/CFS. Interestingly, homologous sequences between EBNA-6 (a latency gene) may induce autoimmunity [143]. Parvovirus B19 infection has been associated with the progression of ME/CFS, with high levels of TNF-α and interferon-gamma (INF-γ) [142]. Viral infections deteriorate the immune system, resulting in a defective immune system and/or autoimmunity, which may involve natural regulatory autoantibodies (AAB) against G-protein-coupled receptors (GPCR), whose presence correlates with fatigue and muscle pain [144]. Many of the symptoms observed in patients with ME/CFS could be explained by the presence of AAB against muscarinic acetylcholine receptors important in vasoconstriction and hypoxemia (M4, M3 AChR) [145], beta2-adrenergic receptors (β2AdR), resulting in neurocognitive symptoms, pain and endothelial dysfunction [140]. Considerate research is being conducted on the role of gut microbiota in ME/CFS, enteric dysbiosis, and increased gut permeability [146]. Additionally, ME/CFS patients have other immune or autoimmune diseases, such as fibromyalgia hypothyroidism, Hashimoto, and Sjögren’s syndrome [147]. The immune dysregulation and autoantibodies are seen in ME/CFS; particularly important is the association between ME/CFS and the high frequency of autoimmune diseases [148]. The presence of EVs [149] and autoantibodies have been observed in ME/CFS [142]. ME/CFS is currently determined by following the Canadian Consensus Criteria (CCC), Fukuda criteria, International Consensus Criteria (ICC), and the Institute of Medicine Criteria; nonetheless, misdiagnosed, and undiagnosed patients represent a large population due to the complexity of the condition [150].

Specific diagnostic molecular markers remain to be elucidated for ME/CFS [151]. Machine learning classifiers have identified 20 proteins including CSF2, TNF-α unique to ME/CFS versus controls [152]. In addition, CCL11 (Eotaxin-1), chemokine (C-X-C motif) ligand 1 (CXCL1), CXCL10 (IP-10), IFN-γ, IL-4, IL-5, IL-7, IL-12p70, IL-13, IL-17F, leptin, granulocyte colony-stimulating factor (G-CSF), granulocyte-macrophage colony-stimulating factor (GM-CSF), LIF, nerve growth factor (NGF), stem cell factor (SCF), TGF-β and TGF-α [153] correlate with ME/CFS severity. Several molecules including IL-1α, IL-6 and IL-8 may be biomarkers and severity indicators for ME/CFS [3]. Elevated salivary stress hormone cortisol (evening) is implicated ME/CFS symptoms [3]. Hippocampal involvement and neurocognitive impairments have been reported in ME/CFS [154]. PhIP-Seq and metagenomic studies demonstrated a signature model in patients with severe ME/CFS; a rather large and specific serum antibody response against flagellins (a structural protein of the flagella) of *Lachnospiraceae* (a bacteria found in the gastrointestinal tract that produces short-chain fatty acid (SFAC), which supports immunological tolerance and maintain inflammatory equilibrium) [155]. Not surprisingly, epigenetic changes have been observed in ME/CFS. DNA methylation patterns from peripheral blood mononuclear cells (PBMCs) have been associated with the severity of the disease. Importantly, these changes in the methylome profile were detected in regulatory-active regions of the genome that are known to play a role in metabolic, immune, and inflammatory biological processes [151]. Differences of miRNA expression profiling in a gender-specific manner (i.e., miR-22-3p) [156] were linked to neurological disorders and high levels of mitochondrial DNA (mtDNA) associated with serum EVs [157], and have been observed in ME/CFS only after exercise. Likewise, miRNA levels have also correlated with severity (i.e., miR-150-5p) [158] and could be used as biomarkers to distinguish between ME/CFS and fibromyalgia (i.e., miR-143-3p) [150]. Urine metabolomics analysis showed no difference between ME/CFS and subject controls [159] after extreme post-exertional malaise (PEM).

Although inhibition of proinflammatory TNF-α and IL-6, has been useful in other autoimmune disorders, their benefits on ME/CFS need further investigation [160]. Neuromodulators, in low-dose aripiprazole and methylphenidate, can be effective in treating fatigue, brain fog, sleeping disorders, and PEM. ME/CFS is a complex system, regulation of multiple targets appears to be a reasonable approach. Immunoadsorption has shown clinical improvement in ME/CFS apheresis [161]. The phosphodiesterase 5 inhibitor sildenafil, when used in ME/CFS subjects, showed a considerable improvement in fatigue in patients compared to the placebo group [162].

## 7. Amyotrophic Lateral Sclerosis (ALS)

ALS is a devastating upper motor neuron disease [163] with unknown pathogenesis [164,165] and no effective treatment. As a result, despite efforts to identify unique biomarkers [166,167], there is still no objective way to prognosticate ALS. Typically, the onset of this disease is in late midlife, and is mostly fatal within 3–5 years after the detection of the first symptoms [168,169]. Approximately 90% of ALS cases are sporadic (that is, of unknown etiology), and the remaining 10% are familial. Despite the heterogenous and poorly understood etiology, both types exhibit overlapping pathology and common phenotypes, including protein aggregation and mitochondrial deficiencies [170].

Recently, immune dysregulation has been suspected [171,172,173], with neuroinflammation being invoked in general [174,175,176] and for ALS in particular [177,178,179], especially concerning activation of microglia and astrocytes [180,181]. It is interesting that microglia communicate with [182], and can be activated by [183], mast cells, unique effector immune cells that are ubiquitous in the body and are critical in allergies [184], but also inflammatory processes [185]. Mast cells have also been implicated in the pathogenesis of neuroinflammatory diseases, such as multiple sclerosis, Long-COVID and autism spectrum disorder [16]. In fact, degranulated mast cells were reported to be increased, and were associated with myofibers and motor endplates in the quadriceps muscles of patients with ALS but not in controls [186]. One paper reported that the “mast cell stabilizer” disodium cromoglycate (cromolyn) could reduce ALS-like development in the SOD1^G93A^ mouse model of ALS without affecting mortality [187], but this molecule is almost impossible to enter the brain after oral ingestion and is a very weak inhibitor of mast cells [188]. Instead, the flavonoids luteolin (3,4,5,7-tetraxydroxyflavone) and tetramethoxyluteolin (3,4,5,7-tetramethoxyflavone) is a potent inhibitor of both human microglia [189] and human mast cells [190]. In fact, the structural analogue 7,8-dihydroflavone also mimics brain-derived neurotrophic factor [191], which is line with the suggestion that neurotrophic factors [192] and growth hormone analogues [193] may be useful in ALS.

Over 50 disease-modifying genes have been identified for ALS [194]; mutations in chromosome 9 open reading frame 72 (C9ORF72) [195], CuZn superoxide dismutase type-1 (SOD1) [196,197], TAR DNA-Binding (TARDBP) [198], and fused in sarcoma (FUS) [PMID: 19251627] [199], account for 70% of all cases of fALS. Other genes including p62 (*SQSTM1*), Ubiliquin-2 (*UBQLN2*), TANK-binding kinase 1 (*TBK1*) and Optineurin (*OPTN*) account for less than 1 % each [200]. C9ORF72, SOD1, TARDBP, and FUS genes are key to the normal functioning of motor neurons and other cells. It is unclear how mutation in these genes contribute to the death of motor neurons. Cu/Zn superoxide dismutase (SOD1) was the first causative gene identified to harbor mutations linked to ALS [201]. In total, 20% of the familial cases and 1–2% sporadic ALS are caused by mutations in the Cu/Zn superoxide dismutase (SOD1) gene. The disease is not due to lack of function of the protein, since deleting the gene in animal models does not cause ALS [202]. There appears to be a gain of new toxic function and show increased tendency of mutant SOD1 molecules to aggregate and form clumps in motor neurons and astrocytes. Mutations in TAR DNA binding protein 43 (TDP43) cause a dominant form of ALS, and are responsible for about 4% of familial ALS and about 1% of sporadic ALS [203]. Mutations in the TDP43 gene cause the TDP43 protein to mislocalize in motor neurons in the cytoplasm where it aggregates into clumps. The cytoplasmic mislocalization and aggregation of TDP-43 is a major pathological hallmark for sporadic ALS [204,205]. Similar TDP-43 pathology is found in familial ALS patients with mutation of optineurin, C9ORF72 and VCP, but is not observed in those with mutation of SOD1 and FUS, suggesting that TDP43 may play a pivotal role in many forms of ALS [206]. A hexanucleotide GGGGCC repeat expansion within the C9orf72 gene has been identified in at least 8% of sporadic ALS and more than 40% of familial ALS. This GGGGCC repeat expansion in the noncoding region of the C9ORF72 gene is the most common genetic abnormality in familial and sporadic ALS- and frontotemporal dementia (FTD) [203,204]. Mitochondrial dysfunction is one of the earliest pathophysiological events in ALS and have been thoroughly investigated in familial and sporadic forms of ALS [203,207]. Mitochondrial defects observed in ALS range from excessive ROS production that results in oxidative stress to low energy levels because of inadequate ATP synthesis, defective organelle morphologies, or an imbalance between mitochondrial biogenesis and degradation [208]. Multiple ALS-related genes, such as OPTN, TBK1 and SQSTM1, are directly involved in mitophagy [208]. Mitochondria in ALS linked SOD1 mutant model experience axonal transport deficits before symptoms occur, resulting in a deficiency of axonal mitochondria [209]. Mutant SOD1 preferentially binds to mitochondria, thus interfering with the OXPHOS process and resulting in damaged mitochondria, as well as disrupted mitochondrial transportation [210]. Both TDP-43 overexpression and mutation decreased mitochondrial length and density [211]. The dipeptide repeats (DPRs) resulting from the hexanucleotide repeat expansion in C9ORF72 disrupt mitochondrial function by altering IMS proteostasis and inner membrane architecture [212]. Poly (GR), which results from repeat expansion in C9ORF72, favorably binds to mitochondrial ribosomal protein and compromises mitochondria function, which leads to increased oxidative stress in neurons [213]. Mitochondrial accumulation has been found in the soma of motoneurons in the spinal cord of ALS patients [208,214]. Accumulation of damaged mitochondria may cause neuronal toxicity leading to synaptic dysfunction and neuronal degeneration. Neuroinflammation is a key contributor to motor neuron degeneration and disease progression, and is characterized by reactive microglia and astroglia, infiltrating T lymphocyte and overproduction of inflammatory cytokines. There appears to be crosstalk between motor neurons, astrocytes, and immune cells, including microglia and T-lymphocytes, which are subsequently activated. Degenerating motor neurons and astrocytes release misfolded proteins in ALS, which activate microglia through CD14, TLR 2, TLR4, and scavenger receptor-dependent pathways [180,215]. Anti-inflammatory and neuroprotective factors sustain the early phase of the disease but as the disease progresses, inflammation becomes proinflammatory and neurotoxic.

There has recently been renewed efforts in the development of therapeutics for ALS [216] invoking the potential role of mitochondria [217] and the gut–brain axis [218], as well as anti-sense therapies [219] and personalized immunotherapy [220]. Future efforts to better understand the pathogenesis of ALS and screen for effective treatments will use human organoids composed of iPSC [221,222] discussed in the last section.

## 8. The Use of iPSC-Derived Cells in Neuroinflammatory Disease Pathogenesis and Neurotherapeutics

iPSCs-derived neural cells are used in modeling neuroinflammatory diseases due to their potential to replicate human diseases more accurately than traditional models [223]. hiPSCs are generated from peripheral somatic cells obtained from any individual with or without a diagnosis by reprogramming with a set of core pluripotent transcription factors. Because of their self-renewal characteristic and capacity to differentiate into a wide range of cell types, they are considered a great tool for disease modeling, drug discovery and regenerative therapy [224,225,226]. Figure 3 depicts the differentiation and potential applications of hiPSC in neuroinflammatory disease pathogenesis and therapeutics. Since the genetic background of the donor is retained during cell reprogramming, hiPSCs are an excellent source to analyze the effect of disease-inducing mutations in the CNS cell types such as neurons, glial cells, pericytes, brain microvascular endothelial cells during the disease pathogenesis [227,228]. In addition to the existing efficient monolayer differentiation protocols [229,230,231], more complex multicellular and three-dimensional (3D) organoids have been developed to capture disease-relevant multicellular interactions and closely examine human neuronal development and human brain disease pathogenesis in tissue culture [19,20]. Several patient-specific iPSC lines have been generated modeling and treating various neurodegenerative diseases, including PD, AD, ALS, TBI, ME/CFS, and GWI [232,233,234,235,236,237]. Recently, successful transplantation of iPSC-derived dopaminergic neurons in patients with PD has shown significant improvement in patients’ symptoms and quality of life, leading to potential therapeutic approaches and also for other neurodegenerative diseases such as AD, ALS, Huntington’s disease [238,239]. The autologous cell therapy approach of direct reprogramming of midbrain astrocytes to replace lost dopaminergic neurons in PD has shown a promising alternative therapy for PD [240,241]. The significance of the Gdap1 gene has been recognized for its link to rapid degeneration in iPSC-derived motor neurons due to an altered mitochondrial metabolism-induced redox-inflammatory axis [242]. Cross-comparison of hiPSC motor neurons-derived from familial and sporadic ALS patients by single-cell RNA sequencing revealed early MN-specific disease signatures and identified ELAVL3 misexpression as a new hallmark in ALS therapeutic drug discovery [243]. Elevated levels of tau protein, reduced microtubule acetylation, mitochondrial dysfunction and impaired neurogenesis were observed in iPSC-derived neurons from GWI veterans exposed to the toxicant regimen [236]. Further iPSC-derived cerebral organoids from GWI veterans display cognitive deficits and impaired neurogenesis and could be used to establish GWI toxicants platform for developing personalized medicine approaches for the veterans [244]. Additionally, reactive astrocytes from Multiple Sclerosis patients were found to offer enhanced axonal protection under inflammatory conditions, indicating the therapeutic potential of modulating astrocyte phenotypes to reduce neuronal damage [245]. Further, the establishment of iPSC-derived BBB models has contributed to the understanding of BBB disorders, and to the study of the earliest stages of BBB dysfunction associated with the disease, which is difficult to ascertain from postmortem tissue [246,247]. A new tri-culture system combining neurons, astrocytes, and microglia from human pluripotent stem cells has been introduced to study neuroinflammation, especially in the context of AD [18]. The inflammatory response of hiPSC-derived microglia from post-TBI was examined, revealing the central role of TNF in triggering downstream cytokine changes [235]. The development of advanced multi-level and high-throughput detection methods enabled downstream analysis and identification of new biomarkers, development of novel diagnostics and personalized therapeutics. Recent evidence suggests that neuromuscular junction (NMJ) is highly vulnerable in ALS, and the defects in NMJ occur before the onset of symptoms [248]. hiPSC-derived neuromuscular organoids provide a reliable model to study the initial cellular pathologies, which is not possible in patients as ALS is diagnosed quite late [249]. Three-dimensional microfluidic NMJ models have been used to gain insight into the morphological and functional readouts at the NMJ contributing to the ALS pathology [250].

iPSC-derived mast cells are pivotal in neuroinflammatory responses, generated through protocols recapitulating the in vivo developmental stages by controlled exposure to growth factors, are crucial in conditions like migraine and multiple sclerosis, where mast cell activation exacerbates CNS inflammation, potentially leading to BBB disruption and immune cell recruitment [251]. Studying these mast cells in vitro allows for insight into their behavior and responses, providing a platform to understand their contributions to neuroinflammatory diseases, identify therapeutic targets, and modulate mast cell activity [251]. Utilizing iPSC-derived cells for neuroinflammatory disease modeling has offered insights into disease mechanisms, with microglia and astrocytes derived from patients highlighting alterations in immune responses, cytokine secretion, and interactions with neurons, elucidating the intricate interplay between immune responses and neurodegenerative processes, presenting potential targets for therapeutic intervention [252]. A few other recent studies have also expanded our understanding of viral infections and the associated neurological complications. iPSC- derived cells can be used as 2D and 3D cultures for CNS disease modeling, phenotypic assays, neural differentiation analysis, functional characterization, and cell migration studies, as well as cell therapies [253,254,255,256,257]. The potential of 3D brain organoids, derived from iPSCs, to replicate the CNS cell networks, shedding light on infections like HIV-1, HSV-1, and SARS-CoV-2 has been described [258]. Further, the establishment of choroid plexus organoids from hiPSCs showcased that SARS-CoV-2 infection led to heightened cell death and inflammatory response [259].

iPSC-derived cells have revolutionized drug screening, enabling high-throughput testing of potential therapeutics by exposing neural cells to compounds and assessing their effects on disease-related phenotypes, expediting the identification of therapeutic compounds [260]. Personalized medicine using patient-specific iPSCs holds promise, as models also recapitulate patient-specific pathologies, enabling testing of potential therapeutics on patient-derived cells, and facilitating the identification of personalized treatment strategies [261]. Despite the great potential of iPSC-derived cellular models, challenges remain, including their immaturity as compared to adult cells, which might hinder the modeling of advanced-stage neurodegenerative diseases, risk of tumorigenicity, potential limitations in replicating complex neural interactions, and the clinical translation of in vitro findings requiring further optimization of the existing differentiation protocols and culture conditions [225,262]. Ongoing research aims to overcome these limitations by refining the differentiation protocols to generate more mature and region-specific cell types incorporating multicellular 3D organoid culture systems for enhanced disease-relevant human-based models [263]. Cutting-edge technologies such as CRISPR/Cas9 gene editing, single-cell RNA sequencing, and microfluidics organoids-on-a-chip culture systems have advanced iPSC-derived cell studies, allowing precise genetic modifications, in-depth assessments of the molecular and cellular heterogeneity, and generation of complex 3D models that enhance research on neuroinflammatory diseases [264]. As multidisciplinary collaborations span across several disciplines like stem cell technology, immunology, neurobiology and regenerative medicine, iPSC-based approaches are poised to reshape neuroinflammatory disease research and therapeutic development.

## 9. Conclusions

Neuroimmune, neuroinflammatory and neurodegeneration processes are implicated in AD, PD and TBI. Chronic neuroimmune dysfunctions associated with neuroinflammation but without apparent neurodegeneration is implicated in both GWI and ME/CFS. Chronic activation of glial cells, brain resident and peripheral immune cells could release several neuroinflammatory cytokines, chemokines and neurotoxic mediators that mediate chronic neuroinflammation, synoptic loss, neurodegeneration and neuronal loss in the brain. Chronic neuroinflammation can affect specific types of neurons in specific regions of the brain and induce specific neuroinflammatory and neurodegenerative disorders. However, there is still a need for more specific biomarkers or signatures, as well as human disease surrogates. iPSC-derived neurons, astrocytes, microglia, endothelial cells, and pericytes could be used for disease modeling and neurotherapeutics for neuroinflammatory diseases.

## Figures and Tables

**Figure 1 cells-13-00511-f001:**
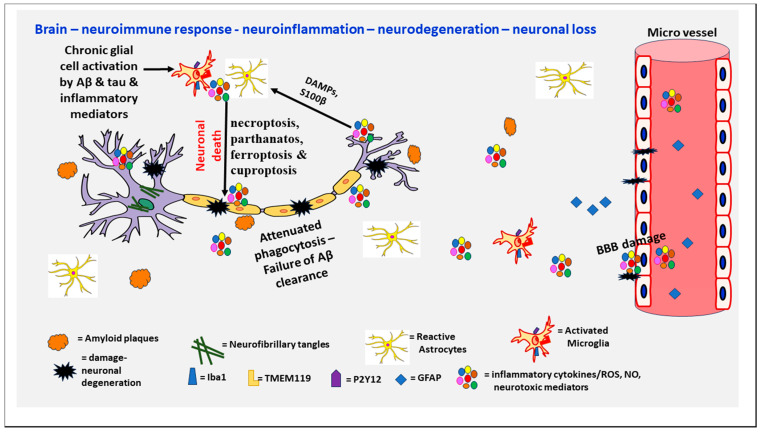
Schematic diagram shows neuroimmune response, neuroinflammation, neurodegeneration and neuronal loss in AD. AD pathogenesis involves neuroimmune response and is associated with the formation of APs and NFTs leading to chronic neuroinflammation, BBB disruption, synaptic loss and neuronal damage in the brain. Chronic activation of astrocytes and microglia release several proinflammatory cytokines, chemokines and neurotoxic mediators (ROS, NO) that further activate glial cells and neurons and induce neuronal death. Molecules released from dying neurons such as DAMPs, S100β, etc. further activate glial cells. Inflammatory cytokines and chemokines from the periphery could enter the brain through damaged BBB and increase neuroinflammation and neurodegeneration. Iba1 = Ionized calcium binding adaptor molecule 1; DAMPS = damage-associated molecular patterns; GFAP = Glial fibrillary acidic protein; NO = nitric oxide; P2Y12R = P2Y12 receptor; ROS = reactive oxygen species; TMEM119 = transmembrane protein 119.

**Figure 2 cells-13-00511-f002:**
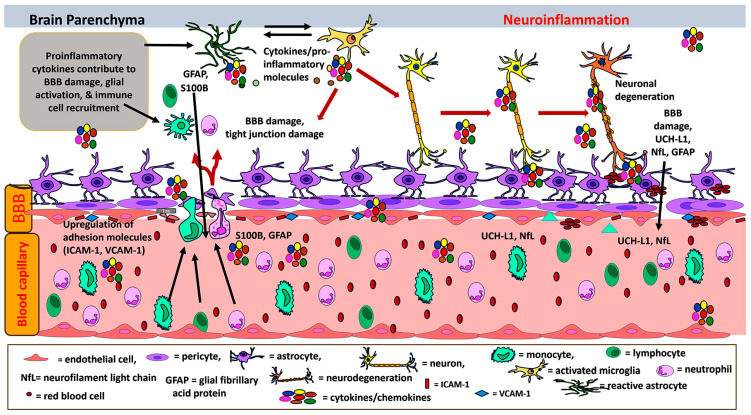
Schematic illustration depicts neuroinflammatory processes in the pathogenesis of TBI. TBI-induced immune response activates microglia, astrocytes and immune cells in the brain, and releases neurotoxic proinflammatory cytokines and chemokines, leading to neuroinflammation, BBB disruption and neuronal death. Primary damage directly causes neuronal injury, whereas secondary brain damage is due to the neuroinflammatory processes following TBI. Inflammatory mediators released from activated glial cells further activate glial cells and release additional neuroinflammatory molecules leading to neuroinflammation, BBB disruption, neurodegeneration and neuronal death. NfL and UCH-L1 are released from damaged neurons and GFAP and S100B are released from activated and damaged astrocytes. NfL, UCH-L1, GFAP and S100B released from brain cells enter into the blood through damaged BBB. BBB = blood–brain barrier; GFAP = glial fibrillary acidic protein; ICAM-1 = intercellular adhesion molecule-1; UCH-L1 = ubiquitin C-terminal hydrolase L1; VCAM-1 = vascular adhesion molecule-1.

**Figure 3 cells-13-00511-f003:**
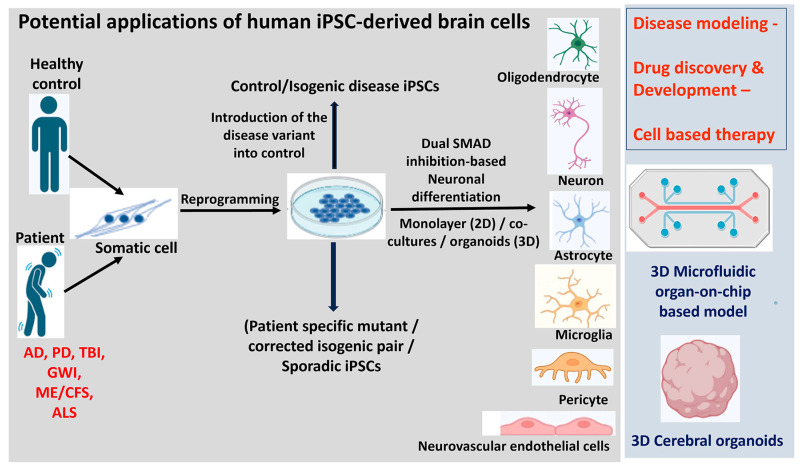
Graphical diagram of the derivation, differentiation, and potential applications of human iPSC in neuroinflammatory disease pathogenesis and therapeutics. iPSCs are derived from somatic cells isolated from patients or healthy donors by reprogramming. CRISPR/Cas9-mediated gene editing is used to correct or introduce the mutation to generate isogenic iPSCs. They can be differentiated into neural precursors using dual SMAD inhibition-based protocols and cultured in monolayer (2D) or in 3D as organoids. Specific morphogens are applied directing the differentiation to the desired cell fate. These disease-relevant cell types including neurons, oligodendrocytes, astrocytes, microglia, pericytes and vascular endothelial cells differentiated from iPSC either patient-derived or isogenic have potential applications in regenerative medicine and disease therapeutics. AD = Alzheimer’s disease; ALS = amyotrophic lateral sclerosis; GWI = Gulf War Illness; iPSC = induced pluripotent stem cells; ME/CFS = myalgic encephalomyelitis/chronic fatigue syndrome; PD = Parkinson’s disease; TBI = traumatic brain injury.

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
