# Peer review of "Recent Research Trends in Neuroinflammatory and Neurodegenerative Disorders"

_cells, 2024, doi:10.3390/cells13060511_

Round 1

Reviewer 1 Report

Comments and Suggestions for Authors

As this is a subject of review, relating neurological impairment and neuroinflammation, I suggest the autors also include an additional chapter: ALS - Amyotrophic Lateral Sclerosis.

At the XXXIV International Symposium on ALS/MND, Basel, Switzerland, this topic was very dominant.

Author Response

Reviewer 1.

Comment: As this is a subject of review, relating neurological impairment and neuroinflammation, I suggest the authors also include an additional chapter: ALS - Amyotrophic Lateral Sclerosis.

Response: We thank Reviewer 1 for suggesting adding an ALS chapter that could add additional interest for the readers. We now have added a new chapter on ALS as suggested.

Reviewer 2 Report

Comments and Suggestions for Authors

Despite intense investigations, effective therapies are not yet available for different neurodegenerative and neuroinflammatory disorders including Alzheimer’s disease, Parkinson’s disease, traumatic brain injury, etc. This manuscript by Cohen et al describes recent progresses in the field of neuroinflammatory and neurodegenerative disorders. It is nice to see the enhanced focus of this review towards the integrity of blood-brain barrier, a pertinent area of research, which has been poorly explored. This is a very thorough and well-presented review with possible underlying mechanisms. Schematics are exciting and nicely drawn.  

Minor:

The quality of figure 2 can be improved by changing brightness/contrast.

Author Response

Reviewer 2.

Comment: Despite intense investigations, effective therapies are not yet available for different neurodegenerative and neuroinflammatory disorders including Alzheimer’s disease, Parkinson’s disease, traumatic brain injury, etc. This manuscript by Cohen et al describes recent progresses in the field of neuroinflammatory and neurodegenerative disorders. It is nice to see the enhanced focus of this review towards the integrity of blood-brain barrier, a pertinent area of research, which has been poorly explored. This is a very thorough and well-presented review with possible underlying mechanisms. Schematics are exciting and nicely drawn.

Minor: The quality of Figure 2 can be improved by changing brightness/contrast.

Response: We thank Reviewer 2 for the comment “This is a very thorough and well-presented review with possible underlying mechanisms” and the suggestion to improve the brightness/contrast of Figure 2. We have now improved the brightness/contrast of Figure 2 as suggested.

Reviewer 3 Report

Comments and Suggestions for Authors

The manuscript (cells-2817916) attempted to review progress on neuroinflammation and neurodegenerative conditions including Alzheimer’s disease (AD), Parkinson’s disease (PD), traumatic brain injury (TBI) and also Gulf War Illness (GWI) and Myalgic encephalomyelitis/chronic fatigue syndrome (ME/CFS) and application of iPSC in mechanistic and therapeutic discovery on these disorders.

Overall, the reviewer finds the review article useful in providing readers an updated summary of recent progress in the field of neuroinflammation, neurodegeneration, neuroinflammatory conditions such as GWI and ME/CFS, and iPSC-related therapy. However, there are a number of errors in the article (see below) that needs to be addressed before publication.

1.      Redundancy in the sentences: “Neuroinflammation may be beneficial or cause detrimental to the brain. Neuroinflammation is characterized by increased levels of inflammatory molecules in the CNS that could cause progressive neurodegeneration and functional impairment. Neuroinflammation may be beneficial or detrimental associated with glial cell activation, and neuronal and brain damage…”; “Chronic inflammation within the CNS has both protective and detrimental roles…”

2.      Ref. 30 and 31 appear to be not correctly cited;

3.      The two sentences appear to be written by different people: “YKL-40 is an astrocytic protein encoded by the gene Chi3l1, and is a cerebrospinal fluid (CSF) biomarker that increases with aging and early in AD.34 Chitinase-3-like protein (CHI3L1/YKL-40), an A1 astrocytic protein, has been identified as another potential CSF biomarker, which increases with aging and early in AD.34

4.      This sentence (line 152-154) does not make sense: “…either studies concentrated on glial fibrillary acidic protein (GFAP) as a marker of astrocytic activation and neuroinflammation”?

5.      “Advances in molecular imaging have enabled further detection of more specific AD markers, such as Colony-Stimulating Factor 1 Receptor (CSF1R) and P2Y12 receptor. CSFR1 is mostly expressed in microglial cells in the brain and contributes to microglial growth, proliferation, and survival.14 Upregulation of this receptor has been shown to parallel with neuropathology in AD. P2Y12 receptors are also microglial receptors that enable the cells to monitor neuronal function. Immunohistochemical staining demonstrated decreased levels of the P2Y12 receptors in the brains of AD patients.14” Original imaging studies of CSFR1 and P2Y12 rather than a review could be cited;

6.      Is this sentence (line 181-182) accurate: “Attempts to treat AD using anti-amyloid strategies did not provide significant benefits”? what’s the reference?

7.      Ref 42 (line 195) is not on AD;

8.       “Other diagnostic modalities such as transcranial sonography, positron emission tomography (PET) scans, or measurement of phosphorylated tau proteins and neurofilament light chains within the CSF may be measured.” Line 219-220 – can the references be provided for p-tau and NfL in PD?

9.      This sentence (line 455-460) needs to be corrected: “One study with GWI mouse models has shown that alteration in immune health restoring bacteria leads to the increased enteric bacteriophage population leads to decreased tight junction proteins resulting in the activation of innate immune mechanisms and toll-like receptor (TLR) signaling pathways which further leads to the increase in serum cytokines such as IL-6, IL-1β, and IFN-γ, intestinal inflammation, and reduced neurogenesis marker, brain-derived neurotrophic factor (BDNF).101

10.  This sentence (line 530-531) does not make sense: “ME/CFS induces or is associated with B cell immune and receptor responses against infection agents.132

Comments on the Quality of English Language

see above.

Author Response

Reviewer 3.

The manuscript (cells-2817916) attempted to review progress on neuroinflammation and neurodegenerative conditions including Alzheimer’s disease (AD), Parkinson’s disease (PD), traumatic brain injury (TBI) and also Gulf War Illness (GWI) and Myalgic encephalomyelitis/chronic fatigue syndrome (ME/CFS) and application of iPSC in mechanistic and therapeutic discovery on these disorders. Overall, the reviewer finds the review article useful in providing readers an updated summary of recent progress in the field of neuroinflammation, neurodegeneration, neuroinflammatory conditions such as GWI and ME/CFS, and iPSC-related therapy. However, there are a number of errors in the article (see below) that needs to be addressed before publication.

Comment 1: Redundancy in the sentences: “Neuroinflammation may be beneficial or cause detrimental to the brain. Neuroinflammation is characterized by increased levels of inflammatory molecules in the CNS that could cause progressive neurodegeneration and functional impairment. Neuroinflammation may be beneficial or detrimental associated with glial cell activation, and neuronal and brain damage…”; “Chronic inflammation within the CNS has both protective and detrimental roles…”

Response: We have edited or removed the sentences indicated for redundancy.

Comment 2: Ref. 30 and 31 appear to be not correctly cited

Response: We thank the Reviewer for this note – we have now changed References 30 and 31.

Comment 3: The two sentences appear to be written by different people: “YKL-40 is an astrocytic protein encoded by the gene Chi3l1, and is a cerebrospinal fluid (CSF) biomarker that increases with aging and early in AD.34 Chitinase-3-like protein (CHI3L1/YKL-40), an A1 astrocytic protein, has been identified as another potential CSF biomarker, which increases with aging and early in AD.34”

Response: We have removed the duplicate sentence and Reference.

Comment 4: This sentence (line 152-154) does not make sense: “…either studies concentrated on glial fibrillary acidic protein (GFAP) as a marker of astrocytic activation and neuroinflammation”?

Response: We have now removed a sentence and fixed it.

Comment 5: “Advances in molecular imaging have enabled further detection of more specific AD markers, such as Colony-Stimulating Factor 1 Receptor (CSF1R) and P2Y12 receptor. CSFR1 is mostly expressed in microglial cells in the brain and contributes to microglial growth, proliferation, and survival.14 Upregulation of this receptor has been shown to parallel with neuropathology in AD. P2Y12 receptors are also microglial receptors that enable the cells to monitor neuronal function. Immunohistochemical staining demonstrated decreased levels of the P2Y12 receptors in the brains of AD patients.14” Original imaging studies of CSFR1 and P2Y12 rather than a review could be cited.

Response: We have added new References but did not find for P2Y12 imaging in humans.

Comment 6: Is this sentence (line 181-182) accurate: “Attempts to treat AD using anti-amyloid strategies did not provide significant benefits”? what’s the reference?

Response: We thank the Reviewer for confirming the statement. Yes, the sentence is accurate and provided the reference for this sentence.

Comment 7: Ref 42 (line 195) is not on AD;

Response: Thank you for the finding. We have replaced it with a correct Reference.

Comment 8: “Other diagnostic modalities such as transcranial sonography, positron emission tomography (PET) scans, or measurement of phosphorylated tau proteins and neurofilament light chains within the CSF may be measured.” Line 219-220 – can the references be provided for p-tau and NfL in PD?

Response: We have removed the whole sentence.

Comment 9: This sentence (line 455-460) needs to be corrected: “One study with GWI mouse models has shown that alteration in immune health restoring bacteria leads to the increased enteric bacteriophage population leads to decreased tight junction proteins resulting in the activation of innate immune mechanisms and toll-like receptor (TLR) signaling pathways which further leads to the increase in serum cytokines such as IL-6, IL-1β, and IFN-γ, intestinal inflammation, and reduced neurogenesis marker, brain-derived neurotrophic factor (BDNF).101”

Response: We have modified this sentence for clarity.

Comment 10: This sentence (line 530-531) does not make sense: “ME/CFS induces or is associated with B cell immune and receptor responses against infection agents.132”

Response: We have modified this sentence for clarity.

Round 2

Reviewer 1 Report

Comments and Suggestions for Authors

Very useful review article of clinical applicability.

The shorter final version was very good.

Reviewer 2 Report

Comments and Suggestions for Authors

It is thoroughly revised.

Reviewer 3 Report

Comments and Suggestions for Authors

I have no more comments.